# Ship-borne lidar measurements showing the progression of the tropical reservoir of volcanic aerosol after the June 1991 Pinatubo eruption.

Juan-Carlos Antuña-Marrero[1], Graham W. Mann[2,3], Philippe Keckhut[4], Sergey Avdyushin[5†], Bruno Nardi[6*] and Larry W. Thomason[7]

[1]Departamento de Física Teórica, Atómica y Óptica, Universidad de Valladolid, 47002, España
[2]School of Earth and Environment, University of Leeds, Leeds, LS2 9JT, UK.
[3]National Centre for Atmospheric Science (NCAS-Climate), University of Leeds, Leeds, UK
[4]Laboratoire Atmosphères, Milieux, Observations Spatiales, Université de Versailles Saint-Quentin, Versailles, 78280, France
[5]Fedorov Institute of Applied Geophysics, Moscow, Russia
[6]Nardi Scientific, LLC, Denver, CO, USA.
[7]NASA Langley Research Center, Hampton, Virginia, USA
[†]Deceased

Correspondence to: Juan-Carlos Antuña-Marrero (antuna@goa.uva.es)

**Abstract:**

A key limitation of volcanic forcing datasets for the Pinatubo period, is the large uncertainty that remains with respect to the extent of the optical depth of the Pinatubo aerosol cloud in the first year after the eruption, the saturation of the SAGE-II instrument restricting it to only be able to measure the upper part of the aerosol cloud in the tropics. Here we report the recovery of stratospheric aerosol measurements from two ship-borne lidars, both of which measured the tropical reservoir of volcanic aerosol produced by the June 1991 Mount Pinatubo eruption. The lidars were on-board two Soviet vessels, each ship crossing the Atlantic, their measurement datasets providing unique observational transects of the Pinatubo cloud across the tropics from Europe to the Caribbean (~40ºN to 8ºN) from July to September 1991 (the Prof Zubov ship) and from Europe to south of the Equator (~40ºN to 8ºS) between January and February 1992 (the Prof Vize ship). Our philosophy with the data recovery is to follow the same algorithms and parameters appearing in the two peer-reviewed articles that presented these datasets in the same issue of GRL in 1993, and here we provide all 48 lidar soundings made from the Prof. Zubov, and 11 of the 20 conducted from the Prof. Vize, ensuring we have reproduced the aerosols backscatter and extinction values in the Figures of those two papers. These original approaches used thermodynamic properties from the CIRA-86 standard atmosphere to derive the molecular backscattering, vertically and temporally constant values applied for the aerosol backscatter to extinction ratio and the correction factor of the aerosols backscattering wavelength dependence. We demonstrate this initial validation of the recovered stratospheric aerosol extinction profiles, providing full details of each dataset in this paper's Supplement S1, the original text files of the backscatter ratio, the calculated aerosols backscatter and extinction profiles. We anticipate the data providing potential new observational case studies for modelling analyses, including a 1-week series of consecutive soundings (in

September 1991) at the same location showing the progression of the entrainment of part of the Pinatubo plume into the upper
troposphere and the formation of an associated cirrus cloud.. The Zubov lidar dataset illustrates how the tropically confined
Pinatubo aerosol cloud transformed from a highly heterogeneous vertical structure in August 1991, maximum aerosol extinction
values around 19 km for the lower layer and 23-24 for the upper layer, to a more homogeneous and deeper reservoir of volcanic
aerosol in September 1991. We encourage modelling groups to consider new analyses of the Pinatubo cloud, comparing to the
recovered datasets, with the potential to increase our understanding of the evolution of the Pinatubo aerosol cloud and its effects.
Data described in this work are available at https://doi.pangaea.de/10.1594/PANGAEA.912770 (Antuña-Marrero  et al., 2020).

## 1. Introduction

Observations by satellite and in situ measurements showed that major volcanic eruptions enhance the stratospheric aerosol layer for several years (Stratospheric Processes and their Role in Climate -SPARC, 2006). Such enhancement causes radiative, thermal, dynamical and chemical perturbations in different regions of the earth's atmosphere, resulting in a perturbation of the earth's climate (e.g. Robock, 2000; Timmreck, 2012). Current research on those perturbations demand detailed information about the 3D spatial and temporal distributions of stratospheric aerosols both under background conditions and after the volcanic eruptions. The June 1991 Mt. Pinatubo eruption is the most used for such research activities because it has been the largest and best documented eruption for the XX century up to the present. Still there are notable gaps in the information collected because the lack of enough measurements but also because several of the measurements conducted and reported in the literature have not been shared by the scientist and institutions that conducted them.

This work is a contribution to the Data Rescue Activity of the Stratospheric Sulfur and its Role in Climate (SSiRC) recently included in this SPARC initiative. This data rescue activity is aimed to "…foster new collaborations between scientists to recover, re-digitize and re-calibrate other historic stratospheric aerosol data sets, and invite scientists to contribute to this activity and to provide advice and expertise on how best to recover other incomplete long term observations of stratospheric composition," (SSiRC, 2020). In its current initial stage particular attention to gather datasets to characterize the progression of the aerosol cloud during the initial months after the 1991 Pinatubo eruption, the main motivation for the work we present here.

Among the envisaged applications of the two Mt Pinatubo's stratospheric aerosols lidar datasets we are presenting is the contribution to future improvements of the Global Space-based Stratospheric Aerosol Climatology, (GloSSAC). GloSSAC is the most complete source of information about the global spatial and temporal distribution of the stratospheric aerosols optical properties from 1979 to the present (Thomasson et al., 2018). From 1979 to mid-2005 the climatology relies mainly on the observations from the Stratospheric Aerosol and Gas Experiment (SAGE) series of satellite instruments. Only two lidar datasets in the tropics were used for filling the gap in SAGE II aerosols extinction profiles in this region in GloSSAC (Thomasson et al., 2018), produced by the dense stratospheric aerosols layer (McCormick and Veiga, 1992).

In section 2 the datasets are briefly described, providing the detailed description, format and inventory of the datasets contained on Supplement S1. Following section 3 describe the processing conducted to try to reproduce the values of the aerosol's extinction profiles at 532 nm for both ship borne lidars Zubov and Vize respectively. Section 4 show and discuss the results comparing them with the available information reported in Avdyushin et al, (1991) and Nardi et al., (1991). The section includes

the discussion of several features of the stratospheric aerosols from Mt. Pinatubo eruption during the period the measurements
were taken to illustrate the importance of the rescued datasets.  Follows section 5 showing an application of the reconstructed
dataset in the validation of Mt Pinatubo modeling simulations. The article conclude with the summary and outlook.

**2. Aerosols Scattering Ratio Datasets**

**2.1 Lidar datasets:**
The single wavelength backscatter measured by a lidar is usually decomposed into two components: aerosol backscatter and
molecular backscatter. The lidar scattering ratio is defined as the ratio between the total backscatter signal (aerosol and
molecular) to the molecular backscatter signal (Collis and Russell, 1976). Here we report the two sets of scattering ratio profiles
measured by two Soviet ship borne lidars a few months after the Mt Pinatubo June 1991 eruption across the north Atlantic
Ocean.  Professor Zubov ship carried a lidar from July to September 1991, and Professor Vize, in January and February 1992
(Avdyushin et al., 1993; Nardi et al., 1993).   The measurements campaign was part of a joint effort between the
Roscomhydromet from the former Soviet Union and the Serviced 'Aeronomie du CNRS* of France.  It included another ship
borne lidar on the French military ship Henry Poincare, based in Brest, and two ground based lidars.  The lidars were located
at the Observatory of Haute-Provence (OHP: 44 °N, 6 °E) and at the Centre d'Essai des Landes at Biscarosse (CEL: 44 °N,
1°W).  A broad description appears in Nardi et al., (1993) and Avdyushin et al., (1993).
Because of the particular spatio temporal distribution of the lidar measurements from Zubov they contribute in characterizing
the variability of the Mt Pinatubo stratospheric aerosols (SA) vertical extinction profiles at certain points and regions of the
North Atlantic Ocean between July and September 1991.  Spatially the variability covers both latitudinal and longitudinal and
temporally the daily variability of two Atlantic locations where lidar measurements were conducted for several consecutive
and nonconsecutive days.
**Table 1: Technical features of the two ship borne lidars. Ya: Yttrium-aluminum.   From table 1 Avdyushin et al., (1991)**

| Lidar Technical Features | Professor Zubov | Professor Vize |
|---|---|---|
| Laser type | Doubled-Ya | Dye:R6W |
| Wavelength (nm) | 539.5 | 589 |
| Energy/pulse (J) | 0.2 | 0.4 |
| Frequency ($s^{-1}$) | 25 | 5 |
| Power (W) | 5 | 2 |
| Emitted Beam Width (rad) | $5 \times 10^{-4}$ | $5 \times 10^{-4}$ |

| | | |
|---|---|---|
| Receiver telescope diameter (cm) | 110 | 110 |
| Filter FWHH (nm) | 0.5 | 0.8 |
| Vertical resolution (m) | 150 | 300 |


## 2.2 Data source

Prof Philippe Keckhut contributed the lidar scattering ratios (SR) profiles dataset derived from the lidar measurements
conducted by Zubov and Vize vessels for the PhD dissertation research of the lead author in 1999. The goal of that research
was to validate the Mt Pinatubo SA extinction profiles measured by the Stratospheric Aerosols and Gas Experiment II (SAGE
II) with ground based lidar observations (Antuña et al., 2002; 2003). However, we found very low information to comply with
the proposed goal due to the combination of two facts. Firstly, the SAGE II profiles were truncated above the main core of the
SA layer in the tropics during almost half a year after the June 1991 Mt Pinatubo eruption. It was the result of the elevated
atmospheric opacity produced by the SA (McCormick and Veiga, 1992). Secondly the few coincident vessel's lidar and SAGE
II extinction profiles measurements, because of the coincidence criteria selected (Antuña et al, 2002). The dataset was not used
and remained stored in the lead author archives since then.

## 2. 3 Dataset description

In brief, the datasets consist of 48 data files from the Professor Zubov vessel containing daily profiles of the lidar SR(z) profiles
and 11 lidar SR(z) profiles from Professor Vize vessel. The trajectories of both ships are shown on Figure 1 with the positions
where the lidar measurements were conducted marked with symbols. The Professor Zubov vessel (red stars) began its
measurement on July 12[th] 1991 from (39 ºN, 28 ºW), travelling towards the Caribbean. After arriving in the Caribbean near
Punta de Maisí (the easternmost point of Cuba), for the last week of July and first weeks of August its trajectory consisted of a
loop around the lesser Antilles island group (see Figure S2), the most southward lidar measurement on August 9[th] (10 N) near
to Trinidad and Tobago. From August 19[th] the Zubov began an eastward trans-Atlantic leg travelling from (21 ºN, 63 ºW) in
the direction of north Africa, 5 co-located lidar measurements made whilst the ship remained for 7 days (September 3[rd] to 9[th])
at its most southward point in the vicinity 8 ºN and 24 ºW . Nine further measurements were made as the ship travelled
northeast towards Europe, the last measurement taken on September 21[st] in the vicinity of the northern Spain.
Whereas the July to September Zubov lidar measurements of the Pinatubo cloud from the Caribbean and Atlantic provide
information on the early stages of the Pinatubo aerosol cloud as it was in transition from its initial sheared plume structure, the
Professor Vize measurements (blue diamonds) were after a substantial proportion of the tropical reservoir of volcanic aerosol
(e.g. Grant et al., 1996) had already been transported to mid-latitudes. The Vize began in the Southern Hemisphere at
on January 26th 1992 (-8ºS, 2º W), moving northward, measuring this later phase of the tropical Pinatubo aerosol reservoir, the
datasets providing a transect of 7 tropical lidar profiles along the western coast of central and northern Africa in the latitude
range 10ºS to 20ºN. from January 26th to February 1st.  The final 4 measurements were then of the mid-latitude Pinatubo cloud,
from 34ºS from just north of the Canary islands, then off the coast of northern Spain, with the final two measurements in the
Baltic sea on February 19th and 20th at 56º and 59 ºN (18º and 27 º E). It should be noted that the Vize lidar dataset contains
only the 11 of the 20 measurements in the two papers, another 9 lidar profiles reported to have been conducted (Avdyushin et
al., 1993; Nardi et al., 1993).

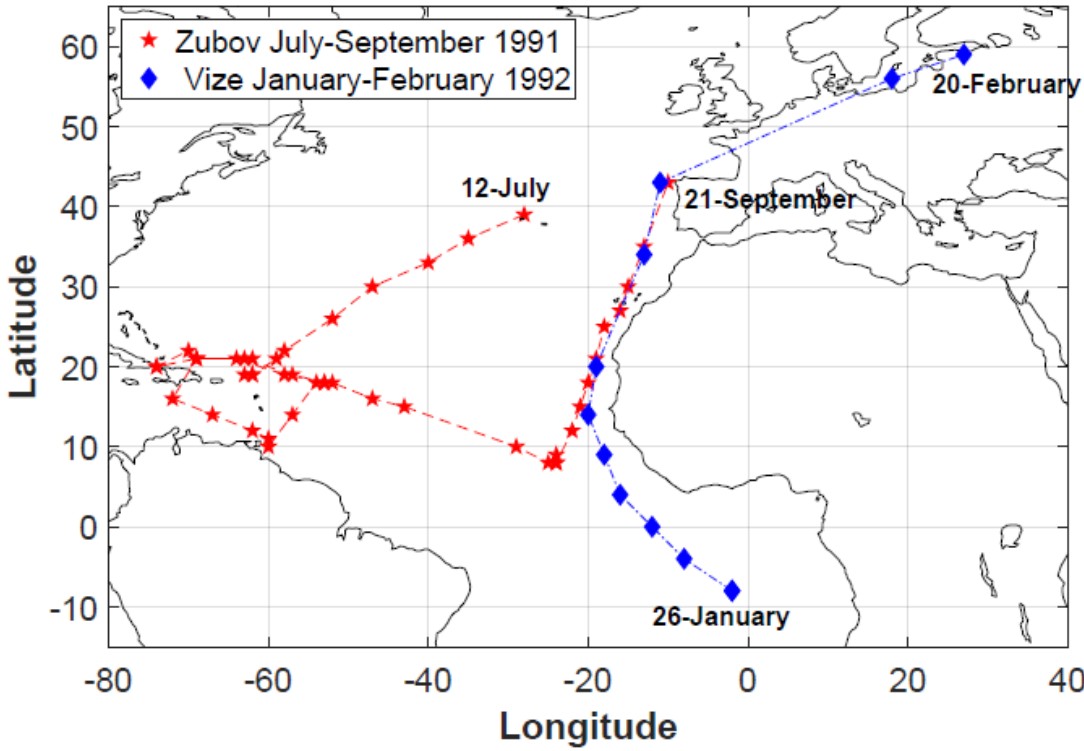

*Figure 1: Trajectories of the Professor Zubov (red stars) between July 12th and September 21st 1991 and*
*Professor Vize (blue diamonds) between January 26th and February 20th 1992.*
**3. Data processing**
To comply with the goal to reproduce the aerosols extinction vertical profiles ($\alpha_{ext}(z)$) reported in Avdyushin et al., (1993) and
Nardi et al., (1993) from the available SR(z), we deliberately followed exactly the same algorithms and parameter assumptions
used in those papers.  This section describes each of the processing steps they conducted, and which we have followed exactly
for the recovered dataset.  To derive the 532nm aerosol signal, the approach taken in both datasets was to specify a Rayleigh
backscattering cross section coefficient of $5.7 \times 10^{-32}$ m$^2$ sr$^{-1}$ at 532 nm.  For the 539nm lidar SR in the Zubov dataset, no
wavelength dependence was accounted for, the wavelength difference from the target 532 nm considered negligible, whereas
for the 589nm lidar SR on the Vize dataset, a correction factor of the Rayleigh backscattering cross section coefficient at 532
nm ($589^{-4}/532^{-4} = 0.666$) was used (Avdyushin et al., 1991).
Then Rayleigh backscatter at the surface was calculated and for each lidar measurement the Rayleigh backscatter profiles
($\beta mol(z)$) were derived using the vertical profiles of pressure ($P(z)$), and temperature ($T(z)$) from the CIRA-86 atmospheric
model (Flemming et al., 1988). The procedure consisted in determining the geopotential height ($Zg(z)$) and $T(z)$ at the
mandatory $P(Z)$ levels from 1000 to 0.1 hPa from the CIRA-86 atmosphere taking into account the month the measurement
was conducted and latitude of the ship for each individual measurement. Then the $Zg(z)$ were converted to geometric altitude
($z$). Following the $P(z)$ were logarithmically interpolated in the vertical to the altitude of the lidar SR levels. Similar step was
conducted for $T(z)$ but using lineal interpolation. Then the $\beta_{mol}(z)$ is derived using the standard procedure (Bucholtz, 1995).
Following the aerosols backscattering profiles ($\beta_{aer}(z)$) were derived using equation 1 (Russell et al., 1979). To avoid cero or
negative values in $\beta_{aer}(z)$, produced by SR(z) equal or lower than 1 respectively, we replaced those SR(z) values by 1.01
following, the value proposed by Russell et al., (1979) for the SR(z) minimum aerosol level. At the levels where this change
took place the magnitude of $\beta_{aer}(z)$ is two orders lower than the magnitude of $\beta_{mol}(z)$ at the same level. Equation 1 was used to
derive $\beta_{aer}(z)$:
$$\beta_{aer}(z) = [SR(z) - 1] \times \beta_{mol}(z) \quad (1)$$
The next step consisted in calculating the $\alpha_{aer}(z)$ from the $\beta_{aerl}(z)$ using equation 2, using a constant value in time and altitude
of 0.04 $sr^{-1}$ for the aerosols backscattering to extinction ratio (Advyushin et al., 1991).
$$\alpha_{aer}(z) = \beta_{aer}(z) \left[\frac{\beta_{aer}}{\alpha_{aer}}\right]^{-1} \quad (2)$$
It is worth to mention that it is more common to use the inverse of the term among squared brackets in the former equation,
termed the extinction-to-backscatter lidar ratio or sometimes simply referred to as "the lidar ratio". However, taking into
account the goal of this work, to reproduce exactly these hitherto unavailable data records, the language and terms used in the
two cited papers has been preserved here. In addition, regarding the magnitude of 0.04 $sr^{-1}$ for the backscattering to extinction
ratio (25 sr if the extinction-to-backscatter lidar ratio definition is used), this value taken to be representative of an aqueous
sulphuric acid aerosol cloud with the particle size distribution suitable for this period, 3-9 months after the Pinatubo eruption,
when the effective radius was greatly enhanced compared to background levels (see e.g. Bauman et al., 2003). Vaughan et al.
(1994) showed how the lidar extinction-to-backscatter ratio for aqueous sulphuric acid clouds decreases for larger particles,
with more moderate volcanic aerosol clouds having higher extinction-to-backscatter ratios (see e.g. Prata et al., 2017). For the
1991 Mt Pinatubo eruption, a set of vertical profiles of extinction-to-backscatter lidar ratio values from 355 to 1064 nm were
produced for each month, based on size distribution fits (Jaeger et al., 1995) to balloon-borne optical particle counter
measurements in mid-latitudes (Deshler et al., 1993).  The conversion factors are a function of the time after the eruption and
the altitude, comprising a set of wavelength exponents to convert aerosols backscatter across several wavelengths between 355
to 1064 nm, and also for aerosol extinction (Jäger and Deshler, 2002).   Since the effective radius enhancement after Pinatubo
was much larger in the tropics than in mid-latitudes (see e.g. Russell et al., 1996; Bauman et al., 2003), it remains a potential
future community research effort to produce a recommended Pinatubo lidar extinction-to-backscatter ratio dataset suitable for
the tropics, and for other major eruption periods.
**4. Results**
The tabulated lidar SR profiles and the calculated $\beta_{aer}(z)$ and $\alpha_{aer}(z)$ profiles at the wavelength of 532 nm from both lidars are
available at https://doi.pangaea.de/10.1594/PANGAEA.912770 (Antuña-Marrero et al., 2020).

**4.1 Validation of the reproduced dataset**
No tabulated data is available for the $\alpha_{aer}(z)$ values used in the cited Avdyushin or Nardi's papers, the only published source of
information about the measurements. In addition, the papers do not conduct detailed discussions or mentions of the extinction
relevant features in the Zubov and Vize datasets.   Here we make use of all the available information to conduct a semi-
quantitative validation for the Zubov dataset. In the case of Vize only is possible to conduct a qualitative validation.
Figures 1a and 1b show the temporal/vertical cross section of the $\alpha_{aer}(z)$ measured by the lidars onboard the Professors Zubov
and Vize ships.  The pink discontinuous line on top of the white background in figure 1a is the altitude of tropopause at the
locations the lidar measurements were conducted.  The tropopause altitudes were derived from the ERA-Interim reanalysis
potential vorticity profiles, interpolating to the height levels of the lidar measurements and select the height of the 1.e-5 PV
surface.
Figure 2a shows, the same pattern of the temporal/vertical cross section of the $\alpha_{aer}(z)$ for the entire Zubov trajectory that the
one reported in figure 2 in Avdyushin et al., (1993).  Both Figures are the main semi-quantitative comparison of the results we
present here with those shown in Avdyushin et al. (1993), also validating our method with the few quantitative values reported
in the two papers. The magnitudes of the $\alpha_{aer}(z)$ are in the same order in both figures as it could be seen comparing the scales
of the color bars in the right side of both them.  A careful comparison between the areas painted in red (corresponding to the
highest values of $\alpha_{aer}(z)$) in both figures show a larger area in Avdyushin et al, (1991) figure 2, an indication of slightly lower
values in the values of $\alpha_{aer}(z)$ we reproduced. Moreover, the maximum $\alpha_{aer}(z)$ value in the reproduced dataset is 0.054 km$^{-1}$ at
23.3 km of altitude on August 4$^{th}$ could be appreciated on figure 2a. Avdyushin et al, (1991) reported the maximum at 18 °N
between 23 and 24 km of altitude with an $\alpha_{aer}(z)$ value of 0.08 km$^{-1}$ the same day. All those facts demonstrate the agreement of
the reproduced dataset with the original one.

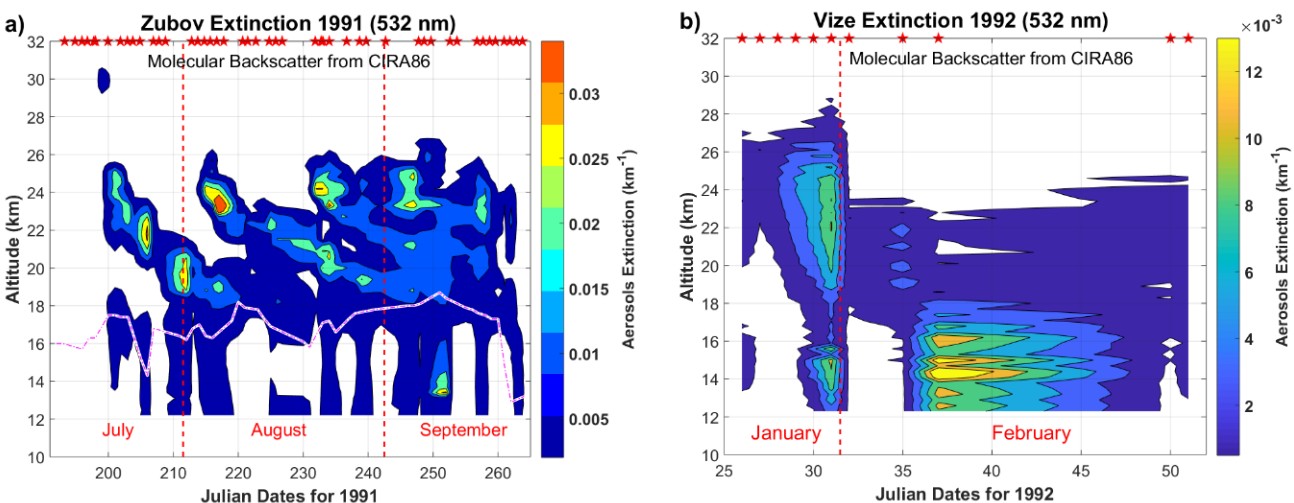


**Figure 2: Temporal/vertical cross sections of the aerosols extinction at 532 nm measured by the lidar onboard the two ship borne**
**lidars during their trajectories. a) Professor Zubov ship; b) Professor Vize ship.**

In the figure 2a it should be also noted the presence of area of high values of the $\alpha_{aer}(z)$ at the tropical middle troposphere in
September 1991 around the day 250. This signature is not seen on the temporal cross section from Zubov lidar on figure 2 in
Avdyushin et al., (1991) because the vertical axes lower altitude is at 15 km. It appears more clearly in the temporal cross
section of the SR(z) from Zubov lidar, the figure 4 in Nardi et al., (1993), having the vertical axes beginning at 12 km. This
feature may be associated to the combination of what seems to be a downward transport of stratospheric aerosols with the
presence of a thick cirrus cloud attached below. The profiles associated to this feature will be discussed later. The features
described above demonstrate that the reproduced $\alpha_{aer}(z)$ dataset in the case of Zubov is in reasonable agreement with the reports
in the only two papers available describing the measurements.
Figure 2b for Prof. Vize shows in general the same pattern than figure 3 in Avdyushin et al., (1993) although the $\alpha_{aer}(z)$
magnitudes in the reproduced dataset are lower. In some way the lack of 9 measurements (~ 45 %) of the 20 reported to be
conducted (Avdyushin et al., 1993) contribute to those low $\alpha_{aer}(z)$ magnitudes in the Vize dataset. Also, in figure 2b the
extension of the vertical axes down to the lower level the lidar information was available, 12 km, allows to see aerosols in the
upper troposphere that is not the case in figure 3 in Avdyushin et al., (1993) figure 3.

**4.2 Downward transport of stratospheric aerosols with a thick cirrus cloud below**

The cited area of high values of $\alpha_{aer}(z)$ at the tropical middle troposphere in September 1991 around the day 250, shown in the
figure 1a is associated to the $\alpha_{aer}(z)$ profile on figure 3 for August $8^h$ 1991. The profile of $\alpha_{aer}(z)$ extents from 24 km in the
lower stratosphere to 12 km, middle/upper tropical troposphere, across the tropopause located at 18.2 km. The most plausible
explanation of the vertical extension of the layer is the occurrence of stratospheric aerosols downward transport into the upper
and middle troposphere. The figure 3 also includes the value of the Total AOD (TAOD) 0.183, resulting from the contributions
of the Stratospheric AOD (SAOD) from the tropopause to 33 km was 0.096 and the upper tropospheric AOD (UTAOD) 0.087,
from 12 km to the tropopause. SAOD and UTAOD have contributions in the same order of magnitudes to the TAOD, showing
the notable magnitude of the stratospheric aerosols into the upper and middle troposphere.
The figure 3 also show that $\alpha_{aer}(z)$ decrease from 0.012 km$^{-1}$ at the 18.2 km (tropopause) up to 0.02 km$^{-1}$ at 17.3 km and then
increases to ending in two sharp maximums at 14 and 13.4 km with $\alpha_{aer}(z)$ of 0.029 and 0.044 km$^{-1}$ respectively. This double
peak layer at the bottom of the Pinatubo stratospheric aerosols layer is a cirrus clouds, a phenomenon already reported for the
Pinatubo. Similar lidar $\beta_{aerl}(z)$ profile structure is reported at Sodankyla (Finland) 66 °N, on figure 1 in Guasta et al., (1994)
for February $3^{rd}$, 1992. This measurement conducted at Sodankyla was part of the European Arctic Stratospheric Ozone
Experiment (EASOE) campaign during the December 1991 to March 1992 where cirrus clouds were reported in 50% of the 56
measurements conducted. Cirrus were reported to grow often within the stratospheric aerosols layer from Mt Pinatubo as in
the case we are discussing (Guasta et al., 1994). This profile shows, probably, the earlier case of a cirrus observed in lidar
measurements of the Mt Pinatubo stratospheric aerosols.
An interesting feature is that in the 48 $\alpha_{aer}(z)$ profiles from the lidar on Professor Zubov vessel between July and September
1991 only in one profile a cirrus cloud was detected, only 2 % of the profiles. However, in 4 of the 11 available $\alpha_{aer}(z)$ profiles
from the lidar on Professor Vize vessel between January and February 1992, 4 profiles showed the presence of cirrus clouds,
around 40% of the observations. These percentage is similar to the reported by a lidar located at Sodankyla, Finland (66 °N),
during the EASOE campaign between December 1991 and March 1992 (Guasta et al., 1994).

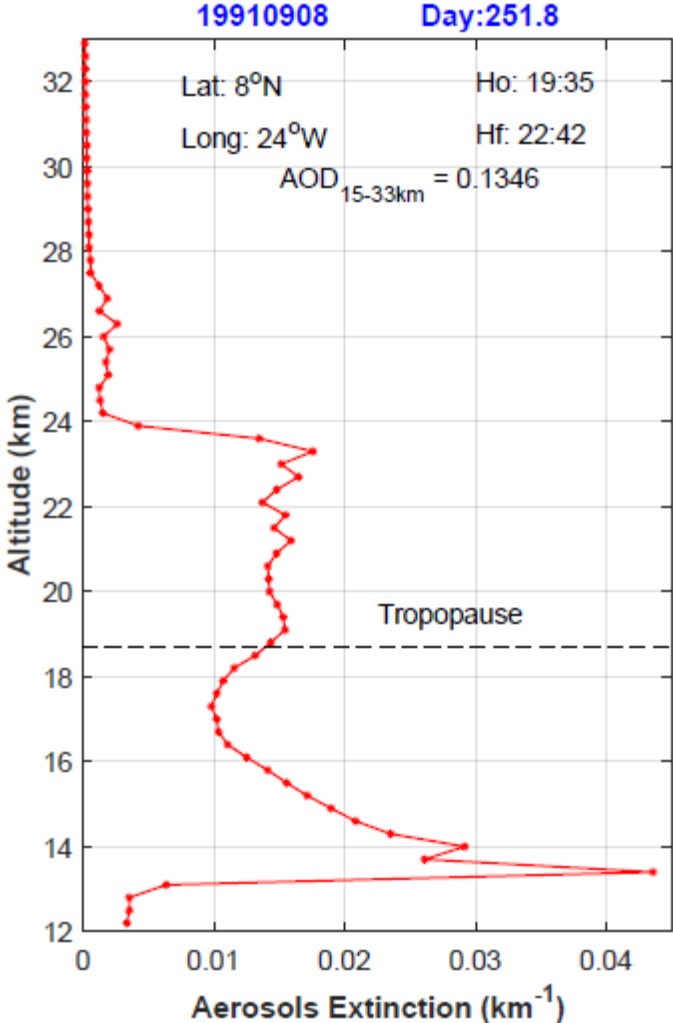


**Figure 3: Profiles of the $\alpha_{aer}(z)$ for September 4th and 5th at 8 °N, showing the presence of cirrus clouds between 13 and 14 km. In**
**addition, the right panel show the transport of stratospheric aerosols from the stratosphere into troposphere across the tropopause.**

**4.3 Absolute maximum $\alpha_{aer}(z)$ value:**
Figures 4a and b shows the $\alpha_{aer}(z)$ profiles on August 3rd and 4th 1991, the figure 4b belonging to the day the absolute maximum
value of $\alpha_{aer}(z)$ was registered and the figure 4a to the day before. Both profiles were taken at the same latitude and only 1°
apart in longitude, allowing to characterize the longitudinal evolution of the Mt. Pinatubo stratospheric aerosols evolution and
variability. A double layer is present both days. The UTAOD is almost the same for both days but SAOD in one order of
magnitude from 0.081 on August 3rd, 1991 to 0.119 the next day.

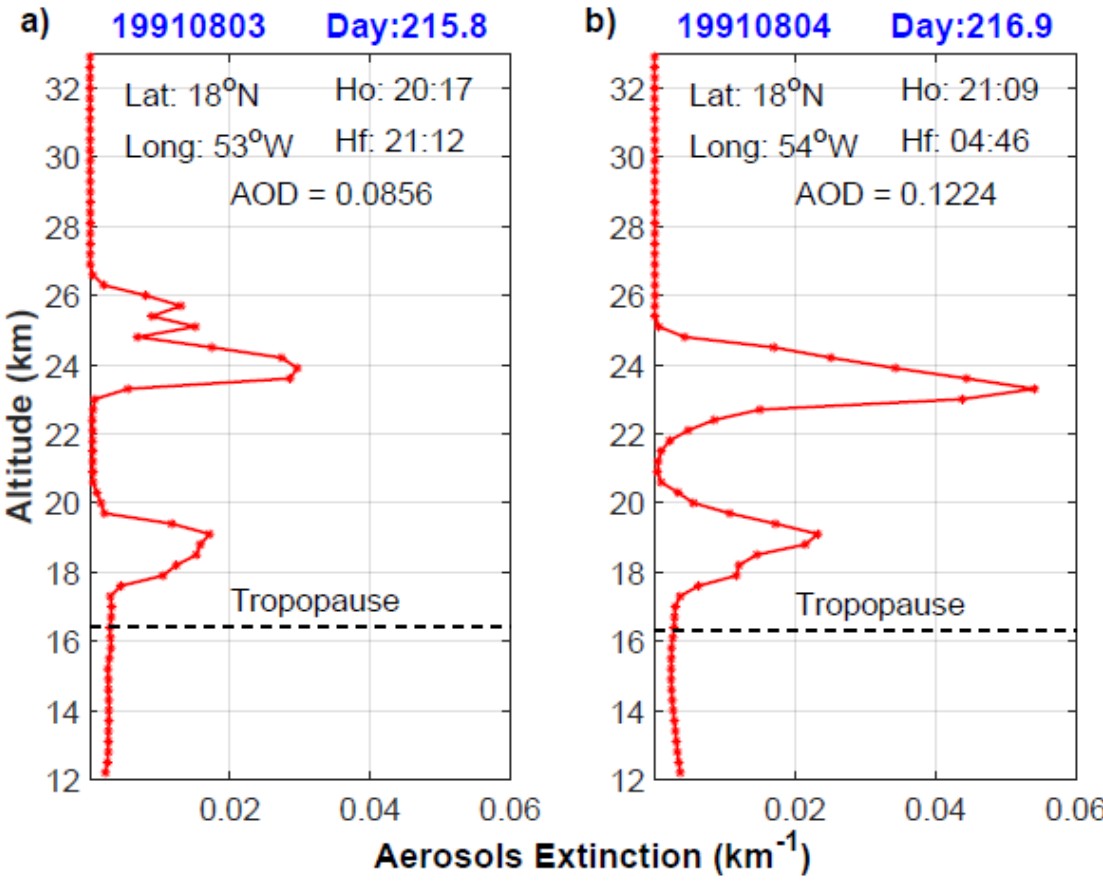


**Figure 4: Profiles of the $\alpha_{aer}(z)$ for August 3$^{rd}$ and 4$^{th}$ at 18 °N.**


On table 2 the geometrical and optical parameters of the higher and lower layers present in both the August 3$^{rd}$ and 4$^{th}$ $\alpha_{aer}(z)$
profiles.  It could be appreciated the altitude descend of both the higher and lower layers from August 3$^{rd}$ to 4$^{th}$, with both layers
keeping their depths. The altitude of the $\alpha_{aer}(z)$ absolute maximum in the top layer decreased a little more than half a kilometer,
but the maximum in the lower layer maintains its altitude. The magnitudes of the maximums $\alpha_{aer}(z)$ in each layer increase, in
2.45 x 10$^{-2}$ km$^{-1}$ in the upper layer reaching the absolute maximum value of the entire record and in the lower layer in 0.62 x
10$^{-2}$ km$^{-1}$. The AOD increases in 0.028 in the higher layer and 0.023 in the lower.  These is an example of the usefulness of the
rescued dataset allowing to quantify those magnitudes during the early stages of the Mount Pinatubo.
**Table 2. Geometrical and optical parameters of the higher and lower layers present in the August 3$^{rd}$ and 4$^{th}$ $\alpha_{aer}(z)$**
**profiles.**

|  | **HIGHER Layer** | | **LOWER Layer** | |
|---|---|---|---|---|
| **DATE** | **19910803** | **19910804** | **19910803** | **19910804** |
| **Top    [km]** | 26.6 | 25.1 | 20.6 | 20.9 |

| Base   [km] | 23.0 | 21.5 | 16.4 | 16.7 |
|---|---|---|---|---|
| ΔH   [km] | 3.6 | 3.6 | 4.2 | 4.2 |
| AOD | 0.049 | 0.077 | 0.031 | 0.054 |
| Max. $\alpha_{aer}(z)$ [km$^{-1}$] | 2.96 x 10$^{-2}$ | 5.41 x 10$^{-2}$ | 1.71 x 10$^{-2}$ | 2.33 x10$^{-2}$ |
| Max. Height [km] | 29.9 | 29.3 | 19.1 | 19.1 |


The former analysis was based on the assumption that the 1° difference in longitude between the positions of Professor Zubov
lidar on August 3$^{rd}$ and 4$^{th}$ 1991 could be negligible compared to the magnitudes of the lower stratosphere winds transporting
the stratospheric aerosols. To support that assumptions we calculated the mean northward and eastward wind components for
both days in the latitude between 15 and 20 ºN and the longitudes 60 to 40 ºW using the NCEP Reanalysis (Kalnay et al., 1996).
The figure S2 on Supplement S3 shows the profile of the lower stratosphere mean wind components for both days in the
selected area around the two lidar locations. The Figure confirms the northward component was insignificant, with the dominant
easterly flow in the stratosphere at that time. At the altitudes of the two aerosol extinction peaks, 19 and 23 km, the easterly
wind component show values of 54 and 72 km h$^{-1}$, which during the 24 h time difference measurements represent  ~1,300 and
1,700 km displacement respectively. Those displacements compare to only ~110 km (for the 1° difference in longitude at 18
ºN), supporting our assumption.

**4.4 Evolution of the daily AOD, maximum $\alpha_{aer}(z)$ and its altitude along the Zubov trajectory**
Figure 5 shows the temporal evolution, along the entire ship trajectory, of the daily maximum $\alpha_{aer}(z)$, its altitude and the aerosols
optical depth (AOD) calculated between 15 and 33 km.  The three months are denoted as the latitudinal and longitudinal bands
the lidar sampled during the Zubov trajectory. Daily maximum $\alpha_{aer}(z)$ values are mainly in the range between 0.0541 and 5.7 x
10$^{-5}$ km$^{-1}$, with a mean and standard deviations values of 0.018  and 0.013 km$^{-1}$. The altitudes of the maximum $\alpha_{aer}(z)$ values
range between 30.8 and 12.2 km, with mean of 21.8 km and standard deviation of 3.5 km.  The AOD mean value is 0.059 with
a standard deviation of a 0.041, showing its maximum value of 0.149 on September 3$^{rd}$ at 8 ºN and 25 ºE.

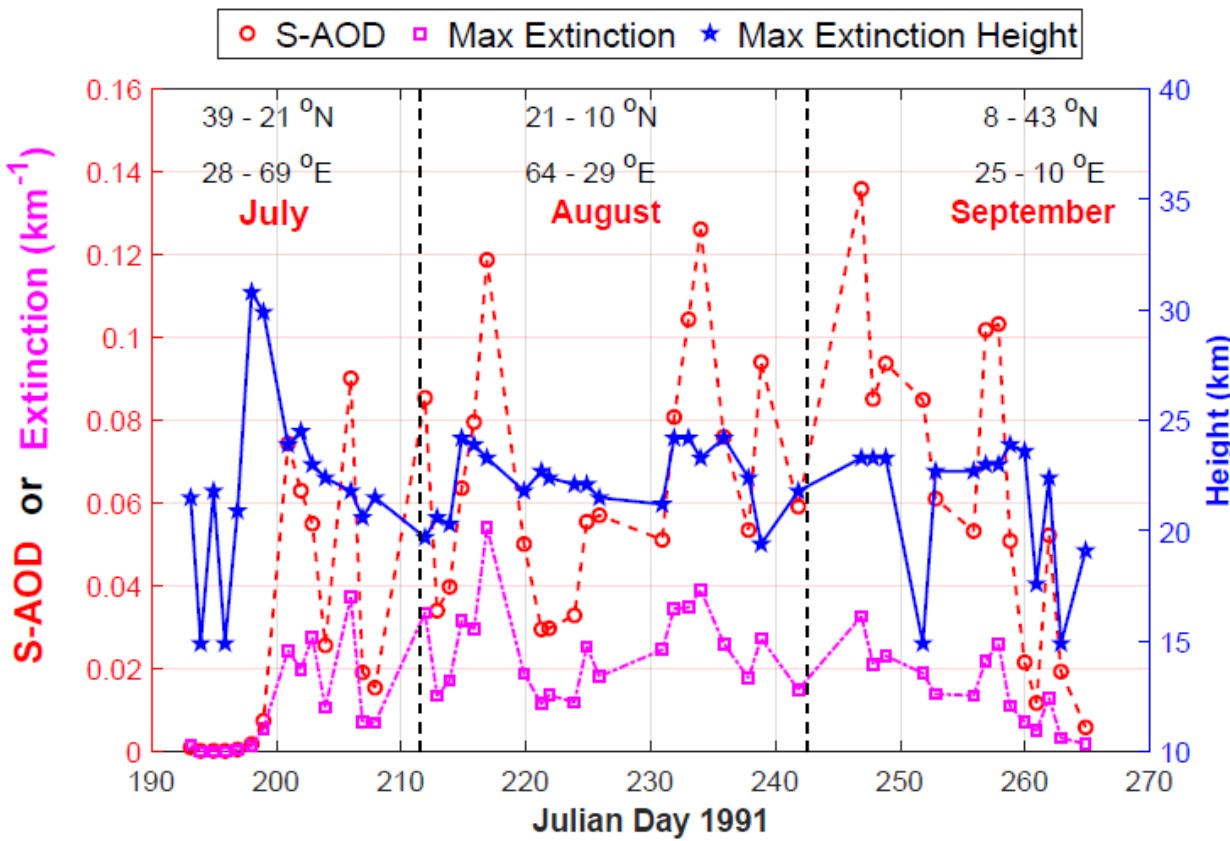


**Figure 5: Temporal section of the AOD, maximum extinction and its altitude from the individual lidar profiles measured by Zubov**

**along its trajectory.**


**5. Data availability**

Data described in this work are available at https://doi.pangaea.de/10.1594/PANGAEA.912770 (Antuña-Marrero et al.,

2020).

**6. Summary and outlook**

Here we present a reproduced version of the stratospheric aerosol extinction profiles derived from lidar measurements

conducted by Professor Zubov and Vize vessels already referenced in the literature (Avdyushin et al., 1993; Nardi et al., 1993)

but unavailable until the present. The data presented consist on two sets of vertical profiles of the SR(z), $\beta_{aer}(z)$ and $\alpha_{aer}(z)$ at

300 m vertical resolution, one for each vessel. In the case of Professor Zubov the set include 48 measurement days conducted
between July and September 1991 and for Professor Vize 11 measurements days between January and February 1992.
We expect this dataset to contribute to some of the current and future research to simulate the early stages of the Mt Pinatubo
eruption. It will also contribute to a future GloSSAC updates, helping to fill the SAGE II gaps produced by the dense
stratospheric aerosols cloud during the first months after the eruption.

**Competing Interest:** The authors declare that they have no conflict of interest.

**Acknowledgements:**
These measurements are the result of the scientific cooperation between Roscomhydromet of the former Soviet Union and the
Serviced d'Aeronomie du CNRS of France and the contributions of the authors of the two cited papers and many anonymous
scientists and supporting people. Despite the social and economic upheaval that occurred with the collapse of the former Soviet
Union, this scientific co-operation between Roscomhydromet and CNRS continued. To both agencies, to the authors of the two
cited papers and the anonymous scientists and supporting staff we recognise the value of this continued collaboration and
express our sincere gratitude to all involved. Juan Carlos Antuña-Marrero acknowledges the support by the Copernicus
Atmospheric Monitoring Service (CAMS), one of six services that form Copernicus, the European Union's Earth observation
programme, for his 1-month visit in March 2019 to the School of Earth and Environment, University of Leeds, Leeds, UK. We
also acknowledge funding from the National Centre for Atmospheric Science for Dr. Graham W. Mann via the volcanic
workpackage of the NERC Multi-Centre Long-Term Science Programme on the North Atlantic climate system (ACSIS). We
also acknowledge discussions, during the CAMS-funded visit to Leeds, with Sarah Shallcross and Sandip Dhomse (Univ.
Leeds) in relation to initial model comparisons to the Zubov lidar dataset. Wind data provided by the NOAA/OAR/ESRL PSL,
Boulder, Colorado, USA, from their Web site at http://psl.noaa.gov/

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
