# Peer review of "Ship-borne lidar measurements showing the progression of the"

_Earth System Science Data, 2020_

## Referee Comment (RC1) · Anonymous Referee #1 · 24 Jun 2020

The authors describe a data set of lidar measurements aboard Russian research vessels over the Atlantic in 1991 and 1992. The data set could be used, e.g. for the evaluation of dispersion modelling of the Pinatubo stratospheric aerosol plume.

The data set is unique and the description is appropriate. The uploaded data sets are reasonable. I only have minor comments that might make it easier for potential users of the data set.

1. The uploaded data sets 3 and 4 (https://doi.pangaea.de/10.1594/PANGAEA.912780

and https://doi.pangaea.de/10.1594/PANGAEA.912781) should be renamed to aerosol backscatter coefficient (rather than aerosol backscattering ratio) to avoid confusion.

2. Please provide a definition of the scattering ratio.

3. It would be useful to provide a plot of the location of the measurements.

4. Please use the extinction-to-backscatter (lidar) ratio in Eq. (2). A value of 25 sr is used here, probably to agree with Advyushin et al. (1991). We now know that stratospheric aerosols from volcanic eruptions have much higher lidar ratios. For instance, Prata et al. (2017, https://doi.org/10.5194/acp-17-8599-2017) find median values around 60 sr at 532 nm while CALIPSO v4 used values between 44 sr and 70 sr (Kim et al., 2018, https://doi.org/10.5194/amt-11-6107-2018). It might be worthwhile to add a brief discussion on more recent findings to put the historic data into perspective.

5. The line marking the tropopause in Figure 1a is pink, not black. I'd also suggest to show the profiles in Figure 1 without temporal interpolation. Just as a column for each measurement time. Is it possible to unify the colour bar?

6. The discussion of Figure 3 and Table 2 (e.g. descending aerosol layer, decrease in layer top height) suggests a stationary measurement for which changes could be related to temporal evolution. What is shown here, however, includes the effect of the change in location. Please revise the discussion accordingly.

7. There is a typo in the legend to Figure 4: Heitgh. Please also provide a description of the figure in the figure caption.

---

## Author Comment (AC1) · 13 Jul 2020

"Ship-borne lidar measurements showing the progression of the tropical reservoir of volcanic aerosol after the June 1991 Pinatubo eruption" by Juan-Carlos Antuña-Marrero et al.

Answers to the Comments from Anonymous Referee # 1:

We thank the reviewer for his comments which contributed to improve the manuscript. Our answers to his comments are detailed below.

[Figure]

1. The uploaded data sets 3 and 4 (https://doi.pangaea.de/10.1594/PANGAEA.912780 and https://doi.pangaea.de/10.1594/PANGAEA.912781 ) should be renamed to aerosol backscatter coefficient (rather than aerosol backscattering ratio) to avoid confusion.

Answer: Data sets 3 and 4 were renamed to aerosol-backscatter-coefficient. https://doi.pangaea.de/10.1594/PANGAEA.912780 https://doi.pangaea.de/10.1594/PANGAEA.912781

2. Please provide a definition of the scattering ratio.

Answer: Text defining the scattering ratio and providing a reference was it was added: Page 4 Line 79:

"The single wavelength backscattering measured by a lidar is usually decomposed into two components: aerosol backscatter and molecular backscatter. The lidar scattering ratio is defined as the ratio between the total backscatter signal (aerosol and molecular) to the molecular backscatter signal (Collis and Russell, 1976)."

The added reference:

Collis, R.T.H. and P.B. Russell, Lidar Measurement of Particles and Gases by Elastic Backscattering and Differential Absorption. In Laser Monitoring of the Atmosphere, E.D. Hinkley, ed. (Springer-Verlag, NewYork 1976), p. 102, 1976.

3. It would be useful to provide a plot of the location of the measurements.

Answer: A plot with the location of the measurements was included in the manuscript, identified as "Figure 1" and a text describing it was it was added.

Page 5 Line 109:

"The trajectories of both ships are shown on Figure 1 with the positions where the lidar measurements were conducted marked with symbols. The Professor Zubov vessel (red stars) began its measurement on July 12th 1991 around 40 °N and 30 °W, travelling towards the Caribbean. Upon reaching the Caribbean, near Punta de Maisí the eastern point of Cuba, by the last week of July its trajectory consisted in loop around the Antilles, except, Cuba. By early August it moved from around 20 °N and 65 °E across the Atlantic in direction to Africa reaching 10 °N and 20 ° E by the first week of September. Then it moved northeast in direction to Europe, conducting it last measurement on September 21st in the vicinity of the northern Spain. A map of the Caribbean loop trajectory is available as Supplement S2. Professor Vize measurements (blue diamonds) began at 0° longitude and -10 °N on January 26th 1991 moving northward, mainly bordering Africa and Europe ending on February 20th around 60 °N and 20 ° E."

Also the Supplement S2 (Attached) was added, consisting a map of the Caribbean Trajectory Loop describing it in detail.

4. Please use the extinction-to-backscatter (lidar) ratio in Eq. (2). A value of 25 sr is used here, probably to agree with Advyushin et al. (1991). We now know that stratospheric aerosols from volcanic eruptions have much higher lidar ratios. For instance, Prata et al. (2017, https://doi.org/10.5194/acp-17-8599-2017) find median values around 60 sr at 532 nm while CALIPSO v4 used values between 44 sr and 70 sr (Kim et al., 2018, https://doi.org/10.5194/amt-11-6107-2018). It might be worthwhile to add a brief discussion on more recent findings to put the historic data into perspective.

Answer: The complete section "3.Data Processing" is devoted to describe the processing Advyushin et al. (1991) reported they conducted. That was the algorithm we repeated to reproduce their results. That is the reason in the Eq. (2) we use the backscattering to extinction ratio, to reproduce exactly their equations and terms.

To reinforce our purpose to provide exactly the equations and terms they used we included on Page 6, line 124:

"The following is a brief description of the steps they conducted and that we followed step by step." Following the suggestion of the reviewer a brief discussion about the

magnitude of the lidar extinction-to-backscatter ratio used in this case. We clarified also the definition of extinction to backscatter lidar ratio.

Page 6 line: 146

"It is worth to mention that it is more common to use the inverse of the term among squared brackets in the former equation, denominated the extinction-to-backscatter lidar ratio. However, taking into account the goal of this work, to reproduce the up to the present unavailable data record, the language and terms used in the two cited papers has been preserved. In addition, regarding the magnitude of 0.04 sr-1 for the backscattering to extinction ratio (25 sr if the extinction-to-backscatter lidar ratio definition is used) it should be noted that for stratospheric aerosols originated from volcanic eruptions higher magnitudes have been reported (Prata et al., 2017). In particular, for the 1991 Mt Pinatubo eruption a set of vertical profiles of extinction-to-backscatter lidar ratio values from 355 to 1064 nm were produced for each month, based on size distribution fits (Jaeger et al., 1995) to balloon-borne optical particle counter measurements (Deshler et al., 1993). The conversion factors are a function of the time after the eruption and the altitude, comprising a set of wavelength exponents to convert aerosols backscatter across several wavelengths between 355 to 1064 nm, and also for aerosol extinction (Jäger and Deshler, 2002)."

The following references were added:

Deshler, T., B. J. Johnson and W. R. Rozier, 'Balloonborne measurements of Pinatubo aerosol during 1991 and 1992 at 41oN: Vertical profiles, size distribution and volatility', Geophys. Res. Lett., 20, 1435-1438, 1993.

Jäger, H., T. Deshler and D. J. Hofmann, 'Midlatitude lidar backscatter conversions based on balloonborne aerosol measurements', Geophys. Res. Lett., 22, 1727-1732, 1995.

Jäger, H. and T. Deshler, 'Lidar backscatter to extinction, mass and area conversions for stratospheric aerosols based on midlatitude balloon borne size distribution measurements, Geophys. Res. Lett., vol. 29, no. 19, 1929, https://doi.org/10.1029/2002GL015609, 2002.

Prata, A. T., Young, S. A., Siems, S. T., and Manton, M. J.: Lidar ratios of stratospheric volcanic ash and sulfate aerosols retrieved from CALIOP measurements, Atmos. Chem. Phys., 17, 8599–8618, https://doi.org/10.5194/acp-17-8599-2017 , 2017

5. The line marking the tropopause in Figure 1a is pink, not black. I'd also suggest to show the profiles in Figure 1 without temporal interpolation. Just as a column for each measurement time. Is it possible to unify the color bar?

Answer: The color of the line marking the tropopause was corrected in the text. The cross sections figures play a crucial role in the visual semi-quantitative validation of the reproduced results, because of the very few quantitative values cited in the two papers cited, the only source of information we have found.

To stress those facts we added the following text on Page 7, line 170:

"Both Figures are the main semi-quantitative comparison of the results we present here with those shown in Avdyushin et al. (1993), also validating our method with the few quantitative values reported in the two papers."

Because of the facts described above it is not possible to plot a profiles instead of the cross sections. The unification of the color bars will make impossible to conduct the visual semi quantitative comparison in the case of the dataset which is changed.

6. The discussion of Figure 3 and Table 2 (e.g. descending aerosol layer, decrease in layer top height) suggests a stationary measurement for which changes could be related to temporal evolution. What is shown here, however, includes the effect of the change in location. Please revise the discussion accordingly.

Answer: The discussion on the former figure 3 (now figure 4) is based in the fact that both measurements were conducted with one day of difference at exactly the same

latitude (18°N) and only 1° difference in longitude. In fact the second measurements on August 4th was conducted 1° west respect the position the day before, what at that latitude represents 110 km. Assuming are broadly known the magnitudes of the eastward wind speed in the tropics we considered unnecessary to support it.

Considering the reviewer suggestion we added the following text on Page 12 Line 244

"The former analysis was based on the assumption that the 1° difference in longitude between the positions of Professor Zubov lidar on August 3rd and 4th 1991 could be negligible compared to the magnitudes of the lower stratosphere winds transporting the stratospheric aerosols. To support that assumptions we calculated the mean northward and eastward wind components for both days in the latitude between 15 and 20 °N and the longitudes 60 to 40 °W using the NCEP Reanalysis (Kalnay et al., 1996). The figure S2 on Supplement S3 shows the profile of the lower stratosphere mean wind components for both days in the selected area around the two lidar locations. The Figure confirms the northward component was insignificant, with the dominant easterly flow at those levels in the stratosphere at that time. At the altitudes of the two aerosol extinction peaks, 19 and 23 km, the easterly wind component show values of 54 and 72 km h-1, which during the 24 h time difference measurements represent ∼1,300 and 1,700 km displacement respectively. Those displacements compare to only ∼110 km (for the 1° difference in longitude at 18 °N), supporting our assumption."

The figure S2 in the Supplement 3 is attached.

The following reference was added:

Kalnay, E., and Coauthors, The NCEP/NCAR 40-Year Reanalysis Project. Bull. Amer. Meteor. Soc., 77, 437–472, https://doi.org/10.1175/1520-0477(1996)077<0437:TNYRP>2.0.CO;2, 1996.

7. There is a typo in the legend to Figure 4: Heitgh. Please also provide a description of the figure in the figure caption.

Answer: The figure 4 was replaced by a new one with the typo corrected. Because of the inclusion of the Figure showing the trajectories along what the measurements were conducted (in answer to comment # 3 former figure 4 is now figure 5.

[Figure]

**Supplement S2:** Trajectory Loop in the Caribbean.

[Figure]

*Figure S1: Professor Zubov loop trajectory in the Caribbean. The location of the measurements are identified by the red star with having nearby its consecutive order number in black. In four cases a second measurement was conducted at one of the initial locations. It was the case for measurements number 10, 15, 16 and 30, denoted by a blue circle around the red star and the number colored in blue. In one case a third measurement was conducted at the same location the number 27, identified by a magenta square around the blue circle having the red star in the middle with the 27 in magenta.*

**Fig. 1.** Figure S1: Zubov Caribbean Loop
Interactive
comment

**Supplement S3:** Mean lower stratospheric winds for August 3rd and 4th 1991.

[Figure]

*Figure S2: Mean lower stratosphere northward and eastward wind components for August 3rd and 4th 1991 in the latitude between 15 and 20 ºN and the longitudes 60 to 40 ºW using the NCEP Reanalysis data.*

**Fig. 2.** Figure S2: Mean lower stratospheric winds for August 3rd and 4th 1991.

---

## Referee Comment (RC2) · Anonymous Referee #2 · 16 Jul 2020

The paper discusses a very old shipborne lidar data set on stratospheric Pinatubo aerosol observations. The data were collected on two Russian research vessels almost 30 years ago, in July-September 1991 and in January-February 1992. The measurements were published in two papers (in GRL 1993).

Why do we now need another paper on this? This question needs to be answered more clearly! I did not get the point.

Now, in this publication, all 48 out of 48 and 11 out of 20 lidar measurement sessions

are re-analyzed. Ok! But the question remains!

Minor revisions are needed.

Details:

Abstract ... formation of an associated cirrus cloud.... This hypothesis on the role of the volcanic particles on cirrus crystal nucleation .... is based on what? ... Are the ash particles favorable INPs? ...or were the sulfuric acid particles responsible for ice nucleation? Sulfuric acid leads to homogeneous ice nucleation. All this remains speculative.

Table 1: Both lidars had a huge receiver mirror (110 cm diameter of the primary mirror). What motivated the Russians to have such big lidars on both ships...? This is just a question! You do not have to answer that in the paper.

Lines 95-96: These personal notes sound strange in a paper... I would avoid ... to mention Prof. Keckhut and ... PhD dissertation of the lead author... Is that information really worthwhile to be mentioned?

Line 118: Did you use CIRA-86 atmospheric profiles here in the re-analysis? I hope not. You probably used 'modern' GDAS or ERA-Interim reanalysis data or ECMWF profiles, I hope?

Line 124: You did not use Russel et al., 1979, right? You used the Fernald (1984) procedure, I hope! Otherwise you have to repeat the re-analysis by using the Fernald (1984) approach.

Line 131: The question on the lidar ratio of 25 sr for 539 or 589 nm... Please have a look into the article of Jager and Deshler (correction paper, GRL 2003). I think, 25 sr is ok for the first phase after the eruption. And later on the lidar ratio increased with decreasing mean or effective size of the sulfuric acid droplets.

Jäger, H. and Deshler, T.: Lidar backscatter to extinction, massand area conversions

for stratospheric aerosols based on mid-latitude balloon-borne size distribution mea-surements, Geophys.Res. Lett., 29, 1929, doi:10.1029/2002GL015609, 2002.

Jäger, H. and Deshler, T.: Correction to "Lidar backscatter to extinc-tion, mass and area conversions for stratospheric aerosols basedon midlatitude bal-loonborne size distribution measurements",Geophys. Res. Lett., 30, 1382, doi:10.1029/2003GL017189,2003.

Line 148-155: If there is agreement, why do you then publish the observations again? I did not get the point.

Figure 1: Would be nice to have an x-axis also in terms of latitude... You need to explain all shown features in the figure caption. To have the explanation in the main text body is not sufficient. The white line...shows what? The color scale is quite poor.

Line 164: Please avoid any speculation. You need a convicing argumentation when it comes to the point: volcanic influence on cirrus. Even Ken Sassen's paper (Science, 1992?) could not explain it. And offered just speculative arguments.

Line 176: day 250 is probably 8 September . . . and not 8 August. . .

Line 184. . .alpha increased. . . not decreased. . .

Line 190: Cirrus and volcanic liquid particles . . .. Even if the volcanic particles would have had an influence on cirrus development, it would be homogeneous freezing, be-cause there is no solid phase. . . and thus there is no chance to distinguish that from the influence of background sulfate particles.

Line 194. . . so if there are only a few cirrus clouds in the volcanic layers. . . the link to volvanic aerosol is not very solid. . .. And meteorological conditions (midlatitudes vs tropics) play a role as well. . .

Figure 2: please explain Ho, Hf, UTS, UT, S in the caption. . .It is just one sentence. . .

Figure 3: similar to Figure 2. . .

Figure 4 results. Are there other tropical lidar observations for comparison? Hawai lidar observations, maybe?

Figure 4 top: . . .Heitgh. . .

---

## Author Comment (AC2) · 25 Jul 2020

We thank the reviewer for his comments which contributed to improve the manuscript. The comments were numbered. Our answers to his comments are detailed below.

The paper discusses a very old shipborne lidar data set on stratospheric Pinatubo aerosol observations. The data were collected on two Russian research vessels almost 30 years ago, in July-September 1991 and in January-February 1992. The measurements were published in two papers (in GRL 1993).

[Figure]

1) Why do we now need another paper on this? This question needs to be answered more clearly! I did not get the point. Now, in this publication, all 48 out of 48 and 11 out of 20 lidar measurement sessions are reanalyzed. Ok! But the question remains!

Answer: Following reviewer suggestion, in line 77 we included the following paragraph:

"Apart from the figures and few magnitudes of stratospheric aerosols extinction reported in the two papers already cited, no other information is available. Those two datasets never were publically available, been absent in the numerous simulations conducted about the climate effects and the evolution of the stratospheric aerosols from the 1991 Mt Pinatubo volcanic eruption. In this paper we make public the two lidars scattering ratios datasets, reconstruct the stratospheric aerosols extinction vertical profiles and produce the stratospheric aerosols backscattering vertical profiles from both lidars by first time."

2) Abstract : : : formation of an associated cirrus cloud: : :. This hypothesis on the role of the volcanic particles on cirrus crystal nucleation : : :. is based on what? : : : Are the ash particles favorable INPs? : : :or were the sulfuric acid particles responsible for ice nucleation? Sulfuric acid leads to homogeneous ice nucleation. All this remains speculative.

Answer: There is a joint answer in relation to all the comments about cirrus clouds at the end of this document.

3) Table 1: Both lidars had a huge receiver mirror (110 cm diameter of the primary mirror). What motivated the Russians to have such big lidars on both ships: : :? This is just a question! You do not have to answer that in the paper.

Answer: It is a big mirror. It contributed to maximize the backscattered laser signal collection, a critical issue considering the contribution to AOD from marine aerosols (on top of the stratospheric AOD) to the two way transmittance attenuation of the signal. The main goal of the lidar onboard Zubov was to measures mesospheric temperature
Interactive
comment

(Nardi et al., 1993). However, there have been several other lidars with mirrors of the diameters in the same order. The lidar at Langley Research Center, NASA, had a mirror of diameter 48 inches 122 cm. The LITE space lidar had 1m diameter mirror. Two French lidars in the 90′s had mirrors with 120 cm the one at Centre d'Essai des Landes at Biscarosse -CEL: 44 °N, 1°W). and 150 cm the one onboard Henri Poincare ship).

4) Lines 95-96: These personal notes sound strange in a paper: : : I would avoid : : : to mention Prof. Keckhut and : : : PhD dissertation of the lead author: : : Is that information really worthwhile to be mentioned?

Answer: It is a common practice in scientific publications to report the origin of the data used and it became more relevant in current times, seeking transparency and repro-ducibility in the reported research. That is more important when a data rescue work is published to explain where the data was found or who contributed with it. In addition we feel compelled to explain why the data was not used in when it was contributed by Prof. Keckhut, a little more than 20 years ago.

5) Line 118: Did you use CIRA-86 atmospheric profiles here in the re-analysis? I hope not. You probably used 'modern' GDAS or ERA-Interim reanalysis data or ECMWF profiles, I hope?

Answer: Yes, we used CIRA-86 and not any other modern reanalysis. As it is stated in the paper our goal was to reconstruct the two stratospheric aerosols extinction datasets. To comply with that goal we followed all the methodological steps the authors mention in their two papers and also used the same parameters (aerosol backscatter-to-aerosol extinction coefficients, wavelength exponent to convert aerosol backscatter from 589 nm to 532 nm and the Rayleigh backscattering coefficient at 532 nm). For determining the molecular backscatter profiles they used the CIRA-86 atmosphere.

6) Line 124: You did not use Russel et al., 1979, right? You used the Fernald (1984) procedure, I hope! Otherwise you have to repeat the re-analysis by using the Fernald

(1984) approach.

Answer: Nardi et al.,(1993) describe how they derived the scattering ratio and normalized it at 40 km or above (scattering ratio = 1.0). In fact the review of that variable in Supplement 4 reveal in the case of Zubov all the profiles at 40.1 km have the value of 1, been in most cases the only value of 1 in the individual profiles. That is procedure described by Russell (1979). We do not know any reason for them to not apply it. In the manuscript we describe how our processing began from those scattering ratio profiles.

7) Line 131: The question on the lidar ratio of 25 sr for 539 or 589 nm: : : Please have a look into the article of Jager and Deshler (correction paper, GRL 2003). I think, 25 sr is ok for the first phase after the eruption. And later on the lidar ratio increased with decreasing mean or effective size of the sulfuric acid droplets. Jäger, H. and Deshler, T.: Lidar backscatter to extinction, mass and area conversions for stratospheric aerosols based on mid-latitude balloon-borne size distribution measurements, Geophys. Res. Lett., 29, 1929, doi:10.1029/2002GL015609, 2002. Jäger, H. and Deshler, T.: Correction to "Lidar backscatter to extinction, mass and area conversions for stratospheric aerosols based on midlatitude balloon borne size distribution measurements", Geophys. Res. Lett., 30, 1382, doi:10.1029/2003GL017189,2003.

Answer: We agree there are better estimates of the extinction to backscatter ratio than the one used by Avdyushin et al., (1993) and Nardi et al., (1993) for processing Zubov and Vize lidars. However, as it have been explained our goal was to reproduce the original aerosol extinction dataset.

8) Line 148-155: If there is agreement, why do you then publish the observations again? I did not get the point.

Answer: The two datasets have not been published before. The figures and few mentions of the stratospheric aerosols extinction magnitudes in the two papers were used to validate the results of the reproduced vertical profiles stratospheric aerosols extinction. We are making public both datasets. Each of then consists of the reproduced

vertical profiles of the stratospheric aerosols extinction by first time (only available in the two cited papers figures and the citation of some of its values); the backscattering ratios (never published before) and the vertical profiles of the aerosols backscatter (never available before).

9) Figure 1: Would be nice to have an x-axis also in terms of latitude: : : You need to explain all shown features in the figure caption. To have the explanation in the main text body is not sufficient. The white line: : :shows what? The color scale is quite poor.

Answer: In answer to reviewer 1 a plot with the location of the measurements was included in the manuscript, identified as "Figure 1" and a text describing it was it was added. Page 5 Line 109: "The trajectories of both ships are depicted on figure1 by the positions where the lidar measurements were conducted. Professor Zubov (red stars) began its measurement on July 12th 1991 around 40 °N and 30 °W, moving to the Caribbean. Upon reaching the Caribbean, near Punta de Maisí the eastern point of Cuba, by the last week of July its trajectory consisted in loop around the Antilles, except, Cuba. By early August it moved from around 20 °N and 65 °E across the Atlantic in direction to Africa reaching10 °N and 20 ° E by the first week of September. Then it moved northeast in direction to Europe, conducting it last measurement on September 21st in the vicinity of the northern Spain. A map of the Caribbean loop trajectory is available as Supplement S2. Professor Vize measurements (blue diamonds) began at 0° longitude and -10 °N on January 26th 1991 moving northward, mainly bordering Africa and Europe ending on February 20th around 60 °N and 20 ° E." Also the Supplement S2 (Attached in the answer to reviewer # 1) was added, consisting a map of the Caribbean Trajectory Loop describing it in detail.

10) Line 164: Please avoid any speculation. You need a convincing argumentation when it comes to the point: volcanic influence on cirrus. Even Ken Sassen's paper (Science, 1992?) could not explain it. And offered just speculative arguments.

Answer: There is a joint answer in relation to all the comments about cirrus clouds at

the end of this document.

11) Line 176: day 250 is probably 8 September : : : and not 8 August: : :

Answer: Corrected. It is September 8th.

12) Line 184: : :alpha increased: : : not decreased: : :

Answer: There was an error in the magnitude assigned for the aerosols extinction at 17.3 km in the manuscript: it is 0.010 km-1 instead of 0.020 km-1. In the profile it is clear that the extinction decrease from 18 to 17.3 km and then increases up to the second maximum at 14 km. The error in the magnitude of the aerosols extinction and 17.3 km was corrected.

13) Line 190: Cirrus and volcanic liquid particles : : :. Even if the volcanic particles would have had an influence on cirrus development, it would be homogeneous freezing, because there is no solid phase: : : and thus there is no chance to distinguish that from the influence of background sulfate particles.

Answer: There is a joint answer in relation to all the comments about cirrus clouds at the end of this document.

14) Line 194: : : so if there are only a few cirrus clouds in the volcanic layers: : : the link to volcanic aerosol is not very solid: : :. And meteorological conditions (midlatitudes vs tropics) play a role as well: : :

Answer: There is a joint answer in relation to all the comments about cirrus clouds at the end of this document.

15) Figure 2: please explain Ho, Hf, UTS, UT, S in the caption: : :It is just one sentence: 16) Figure 3: similar to Figure 2: : :

Answer: The terms Ho and Hf were described in both figure captions. The terms UTS-AOD, UT-AOD and SAOD were also described in the caption of figure 2. The terms UTS-AOD and UT-AOD were eliminated in figure 3 caption, because they do not

contributed to the discussion. Because a figure showing the trajectories along what the measurements were conducted was added to the manuscript (in answer to Reviewer # 1, comment # 3) then former figure 2 and 3 are now figures 3 and 4 respectively

17) Figure 4 results. Are there other tropical lidar observations for comparison? Hawai lidar observations, maybe?

Answer: Yes there are several. We consider it is not necessary to conduct a comparison or discuss them here, because it is not the goal of the manuscript. However, we may refer the reviewer to a PhD Thesis where they are listed as part of a global compilation conducted in 2002. There is a table with all its information, including its respective references. Also a map show the locations of the ground based lidars and the trajectories of the lidars onboard aircrafts and ships: Antuña, Juan Carlos, 2002, Comparison of SAGE II and lidar stratospheric aerosol extinction datasets after the Mt Pinatubo eruption. PhD Thesis, Rutgers University, 91 pp. (Available at: http://rizalls.lib.admu.edu.ph:8080/proquestfil/3066744.pdf)

18) Figure 4 top: : : :Heitgh: : :

Answer: The figure 4 was replaced by a new one with the typo corrected. Because a Figure showing the trajectories along what the measurements were conducted was added to the manuscript (in answer to Reviewer # 1, comment # 3) former figure 4 is now figure 5.

Joint answer to the comments on the cirrus profile showed in the manuscript:

General Answer: are not reporting the study of the potential interaction between cirrus clouds and volcanic aerosols. Any discussion on this subject if completely out of context. We are showing the potential of the information from this profile and the other 4 from Prof. Vize lidar in early 1992, to conduct case studies. We do not speculate, we show facts and call the attention to it to motivate further research.

2) Abstract : : : formation of an associated cirrus cloud: : :. This hypothesis on the

role of the volcanic particles on cirrus crystal nucleation : : :. is based on what? : : : Are the ash particles favorable INPs? : : :or were the sulfuric acid particles responsible for ice nucleation? Sulfuric acid leads to homogeneous ice nucleation. All this remains speculative.

Answer: In the Abstract we changed the expression: ". . . and the formation of an associated cirrus cloud" By ". . . and the detection of a cirrus cloud below it."

10) Line 164: Please avoid any speculation. You need a convincing argumentation when it comes to the point: volcanic influence on cirrus. Even Ken Sassen's paper (Science, 1992?) could not explain it. And offered just speculative arguments.

Answer: The sentence commented by the reviewer is: "This feature may be associated to the combination of what seems to be a downward transport of stratospheric aerosols with the presence of a thick cirrus cloud attached below." This is a fact no an speculation.

13) Line 190: Cirrus and volcanic liquid particles : : :. Even if the volcanic particles would have had an influence on cirrus development, it would be homogeneous freezing, because there is no solid phase: : : and thus there is no chance to distinguish that from the influence of background sulfate particles.

Answer: The sentence commented by the reviewer is: Cirrus were reported to grow often within the stratospheric aerosols layer from Mt Pinatubo as in the case we are discussing (Guasta et al., 1994). This profile shows, probably, the earlier case of a cirrus observed in lidar measurements of the Mt Pinatubo stratospheric aerosols. We cite what is inconcluded in a peer review published paper.

14) Line 194: : : so if there are only a few cirrus clouds in the volcanic layers: : : the link to volcanic aerosol is not very solid: : :. And meteorological conditions (midlatitudes vs tropics) play a role as well: : : The sentence commented by the reviewer is:

Answer: An interesting feature is that in the 48 $\alpha$aer(z) profiles from the lidar on Profes-

sor Zubov vessel between July and September 1991 only in one profile a cirrus cloud was detected, only 2 % of the profiles. However, in 4 of the 11 available $\alpha$aer(z) profiles from the lidar on Professor Vize vessel between January and February 1992, 4 profiles showed the presence of cirrus clouds, around 40% of the observations. These percentage is similar to the reported by a lidar located at Sodankyla, Finland (66 °N), during the EASOE campaign between December 1991 and March 1992 (Guasta et al., 1994). We are reporting a fact, no speculating.